# A Review on the Morphology, Cultivation, Identification, Phytochemistry, and Pharmacology of *Kitagawia praeruptora* (Dunn) Pimenov

**DOI:** 10.3390/molecules28248153

**Published:** 2023-12-18

**Authors:** Qi Wang, Lulu Ding, Ruihong Wang, Zongsuo Liang

**Affiliations:** Zhejiang Province Key Laboratory of Plant Secondary Metabolism and Regulation, College of Life Sciences and Medicine, Zhejiang Sci-Tech University, Hangzhou 310018, China; kimberlyw1206@163.com (Q.W.); llding6@163.com (L.D.)

**Keywords:** *Kitagawia praeruptora*, morphology, ethnopharmacology, cultivation, early bolting, identification, phytochemistry, pharmacology

## Abstract

*Kitagawia praeruptora* (Dunn) Pimenov, commonly known as Qianhu in China, is a widely used folk Chinese herbal medicine. This article reviews its botanical traits, ethnopharmacology, cultivation techniques, identification, phytochemical compositions, and pharmacological effects. Over 70 coumarin compounds, including simple coumarins, pyranocoumarins, and furanocoumarins, have been isolated within this plant. Additionally, *K. praeruptora* contains other components such as flavonoids, fatty acids, benzoic acids, and sterols. This information highlights the importance of utilizing active ingredients and excavating pharmacological effects. With its remarkable versatility, *K. praeruptora* exhibits a wide range of pharmacological effects. It has been found to possess expectorant and bronchodilator properties, cardiovascular protection, antimicrobial and antioxidant activities, anti-tumor effects, and even antidiabetic properties. It is recommended to focus on the development of new drugs that leverage the active ingredients of *K. praeruptora* and explore its potential for new clinical applications and holistic utilization.

## 1. Introduction

*Kitagawia praeruptora* (Dunn) Pimenov (= *Peucedanum praeruptorum* Dunn), also known as “Qianhu” (前胡) in Chinese, belongs to the Apiaceae family and is a perennial herb that is widely utilized in traditional Chinese medicine (TCM) formulations with a domestication history of more than 1500 years [1]. The classification of *Kitagawia* has been a topic of controversy for a long time, and its delineation and morphological delimitation is not well-defined. Despite the creation of *Kitagawia*, numerous scientists still classify certain species as part of *Peucedanum* sensu lato, and *Kitagawia* has been considered a synonym of *Peucedanum* sensu lato [2,3,4,5]. Until 2017, Pimenov had examined the Chinese Apiaceae-type specimens and proposed five new nomenclatural combinations for *Kitagawia* [6]. At present, there are ten known species within *Kitagawia*, with eight species native to China [7]. The dried roots of *K. praeruptora* are recorded as Peucedani Radix in Chinese Pharmacopoeia for reducing fever and resolving phlegm and commonly used to treat various respiratory disorders, including anemopyretic cold, cough with abundant phlegm, and congested chest [8]. Over the past few years, the medicinal benefits and economic worth of *K. praeruptora*, as well as the specific environmental conditions required for its cultivation, combined with excessive human exploitation have made it a highly prized and valuable species in China [9]. It is artificially cultivated for its roots on a large scale in several Southeastern and Mid-China regions including Zhejiang, Anhui, Hubei, and Jiangxi for its unique medicinal value, especially in the authentic production area of Zhejiang Province. *K. praeruptora* was included in the list of one of the famous authentic medicinal materials “Zhebawei” announced by multiple departments in Zhejiang Province in 2018. Chun’an, Lishui, and other places in Zhejiang Province are traditional authentic cultivation areas of *K. praeruptora* and currently have a cultivation area of over 12,000 acres, with an annual output of more than 2000 tons, accounting for 15% of the national total.

We introduce the research achievements from various perspectives in recent years both domestically and internationally as a comprehensive review article. In the beginning, we provide a brief overview of the botanical characteristics and traditional uses. Subsequently, we delve into topics of cultivation and identification methods including early bolting issues. Furthermore, we discuss numerous isolated compounds and pharmacological activities. Lastly, we share our insights regarding the potential future development prospects of *K. praeruptora*. To the best of our knowledge, there has been no comprehensive review of all aspects of *K. praeruptora*. Thus, we aim to enhance the understanding of this neglected plant and provide valuable evidence for the rational use of *K. praeruptora* as a traditional medicine through this paper.

To conduct a comprehensive literature review, we initially defined the review topic. Our search strategy first involved exploring diverse scientific databases such as Google Scholar, PubMed, Springer Link, ScienceDirect, ScienceFinder, Web of Science, Baidu Scholar, and the Chinese National Knowledge Infrastructure. Employing specific keywords and search terms, we narrowed our focus to five key aspects of *K. praeruptora* (*Peucedanum praeruptorum*): morphological characteristics, cultivation methods, physicochemical identification, compounds isolation, and pharmacology. Furthermore, we set a specific time frame for the literature search, extending up to the end of February 2023. Furthermore, the criteria guiding the inclusion and exclusion of studies encompass several factors: relevance to the research topic, academic credibility with a preference for literature from peer-reviewed academic journals, conference papers, professional books, and methodological diversity. Specifically, we sought diversity in chemical composition and pharmacological effects, including variations in compound isolation methods, animal modeling, and cytokine detection. Finally, we ensured the exclusion of multiple versions of the same study in our review.

## 2. Botany

*K. praeruptora* is a perennial herb that grows up to 1 m in height and has a solitary stem that branches above with puberulous branchlets. The basal leaf blade is triangular-ovate and 2–3-ternate, with long-petiolulate pinnae and pinnules that are long-ovate, 3–5-lobed, and measure 1.5–6 × 1.2–4 cm (Figure 1). Both surfaces of the leaves are glabrous, occasionally puberulous, with prominent abaxial nerves, a cuneate base, irregular serrate margins, and an acuminate apex. The 8–12 ovate-lanceolate bracteoles are shorter than the flowers and have rough-puberulous surfaces, with 15–20-flowered umbellules. The calyx teeth are obsolete, the petals are white, and the styles are short. The fruit is ovoid and measures approximately 4 × 3 mm, with sparse pubescence. The root is conical with forked end. The entire growth cycle of *K. praeruptora* can be categorized into three stages: seedling stage, vegetative growth stage, and reproductive growth stage [10]. During the period of reproductive growth, the pregnancy bud stage is generally around mid-June, the flower time is about two months long, usually from August to September, and the fruit time is from October to November [3]. Wild populations of *K. praeruptora* are widely distributed throughout Central and Eastern regions of China, westward to Sichuan and Guizhou, stretching northward to Henan and Gansu, as well as southward to Guangxi with the preference for habitat of forest margins and grassy slopes at the altitude of 250–2000 m [11] (Figure 1E). Northwestern Zhejiang, southeastern Anhui, and northeastern Jiangxi are authentic production areas of *K. praeruptora*. Root of *K. praeruptora* produced in Anhui and Zhejiang is commonly known as “Ningqianhu” and “Chunqianhu”, which accounts for about 80% of the market share of Chinese medicinal materials. The root of *K. praeruptora* produced in Hubei, Hunan, Guizhou, and other places is called “Xinqianhu”, which accounts for about 20% of the market share [10]. 

## 3. Ethnobotany

*K. praeruptora* is a traditional medicine commonly used in China with a long history of application and has been recorded in many ancient books and modern pharmacopeia [12]. It was first recorded in Ming Yi Bie Lu and Ben Cao Jing Ji Zhu, resolving sputum and relieving asthma. During the Tang and Song dynasties, it was described in many other medical books, such as Xin Xiu Ben Cao, Ben Cao Tu Jing, and Zheng Lei Ben Cao, which were mainly used for the treatment of making expectoration easy. In the Ming and Qing dynasties, it was used for the treatment of typhoid fever, which was recorded in Ben Cao Meng Quan and Ben Cao Feng Yuan. In Ben Cao Gang Mu, a classic book written by Shizhen Li, it was recorded to treat vomiting and regurgitation as well as reducing fever and resolving phlegm.

## 4. Cultivation

*K. praeruptora* is suitable for a cool and moist climate and possesses biological characteristics such as cold resistance, heat resistance, drought resistance, barren soil resistance, and disease and insect resistance [13]. For a long time, materials of *K. praeruptora* medicinal herbs have relied on wild resources collection without classification of commodity specifications and grades, and non-renewable harvesting has led to a sharp decline in the reserves of wild *K. praeruptora* resources [14]. Due to its high medicinal value and large market demand, the cultivation area of *K. praeruptora* is constantly increasing. Currently, there are mainly four types of cultivation modes for *K. praeruptora*, including wild planting, simulated wild planting on barren slopes, planting on newly opened forest land, and field planting [15]. The cultivation technology research focusing on *K. praeruptora* site selection and land preparation, breeding and planting, field management, harvesting and processing, and storage have attracted the attention of many scholars [16]. Current studies proved that the cultivation mode has the greatest impact on the content of coumarin of *K. praeruptora* [17]. *K. praeruptora* grown in wild environments such as valley streams or sunny sparse forest was reported to take several years to produce flowering stems, but artificially cultivated *K. praeruptora* can produce flowering stems in the second year [18]. 

In recent years, the planting scale of *K. praeruptora* has been continuously expanding nationwide; the problems of early bolting, flowering, lignification, and the content of active ingredients in *K. praeruptora* below standard greatly affect the production of *K. praeruptora* [19]. Early bolting refers to the process of transitioning from vegetative growth to reproductive growth by blooming before the vegetative mass is fully developed. Early bolting has always been a key factor that severely restricts the production and industrialization of Apiaceae medicinal herbs, including *Angelica dahurica* (Fisch. ex Hoffm.) Benth. et Hook. f. ex Franch. e, *Saposhnikovia divaricata* (Turcz.) Schischk. [20], *Angelica sinensis* (Oliv.) Diels [21,22], and *K. praeruptora* [23]. During early bolting, endogenous hormones will change to regulate growth and development, which has a greater impact on the root structure and coumarin content of *K. praeruptora*. After bolting, the cross-sectional area of secondary xylem in the root is greatly increased, and there are many wood fibers and vessels (Figure 2). The pericycle and thin-walled cross-sectional area of tissue and phloem decrease sharply, and the content of coumarin components decrease significantly [24]. The roots of *K. praeruptora* that have gone through early bolting often lose their medicinal efficacy and cannot be used as medicinal materials because the content of coumarin components is greatly reduced and does not meet the standards of pharmacopoeia [25]. Therefore, due to the influence of genetics, ecological environment, and other factors, the vegetative growth and reproductive growth of different individuals in the same period are inconsistent, and early bolting has become one of the key issues that seriously restrict the production of traditional Chinese medicine. Effective control of bolting can significantly improve the medicinal quality and yield of *K. praeruptora*.

To decrease the rate of early bolting and increase the content of effective ingredients of *K. praeruptora*, high altitude (900–1200 m) and shading degree of 60% was suitable for *K. praeruptora* [26]. The change of altitude gradient in the same area will cause changes in various environmental factors such as light, humidity, and temperature and directly affect the physiological and biochemical structure and quality. There was a big difference in yield of *K. praeruptora* through different treatment methods; the highest yield of roots was achieved by using silver film covering and float bed seedling technology, and the highest net income was achieved by using silver film covering alone [19]. Many field management experiments mentioned that topping and pruning 3–4 times a year can reduce the occurrence of flowering stems and increase yield, but there are no specific data to support this in the literature.

Most current studies were mainly focused on cruciferous vegetables and large vegetables with straw properties, such as *Brassica oleracea* L. [27], *Allium cepa* L. [28], and *Brassica rapa* L. [29]. The phenomenon of bolting is primarily a quantitative trait that is controlled by multiple genes and regulated by various pathways of environmental signals and endogenous factors [30]. Numerous molecular markers or quantitative trait locus (QTL) loci [31] linked to bolting and flowering traits have been discovered in cruciferous vegetables, and there is a considerable collinearity in genome arrangement with significant variations in the copy number and gene structure of homologous genes [32,33]. Consequently, the genetics and molecular regulatory mechanisms of bolting and flowering in cruciferous crops have always been a hot topic. The process of flowering induction of the model plant *Arabidopsis thaliana* (L.) Heynh. has gradually become clear (Figure 3). It contains several key genes that control flowering [34,35]: CONSTANS (CO), LEAFY (LFY), FRIGIDA (FRI), FLOWERING LOCUS C (FLC), and VERNALIZAION (VRN). But there is still a lack of systematic study in this area. At present, this study has confirmed that some exogenous hormones have a certain inhibitory effect on the early bolting of plants and at the same time can improve the quality of medicinal materials, which has been confirmed in plants such as *A. sinensis* [36], *A. dahurica* [37], and *B. rapa* [38]. In the study of genes related to bolting in *K. praeruptora*, researchers employed a combination of transcriptome sequencing and co-expression analysis to elucidate the regulatory mechanisms of key genes involved in coumarin synthesis and the formation of other active ingredients before and after bolting [39,40]. The results revealed that key genes in the phenylpropanoid pathway (4-coumarate-CoA ligase (4CL) and hydroxycinnamoyl transferase (HCT)), ATP-binding cassette (ABC) transporters, apoptosis-related genes, and genes involved in circadian rhythm regulation (cryptochrome circadian regulator (CRY), GIGANTEA (GI), phytochrome A (PHYA), and phytochrome-interacting factors 3 (PIF3)) may play crucial roles in regulation.

## 5. Identification

At present, the main identification techniques of traditional Chinese medicine include sensory evaluation, microscopic identification [41], physical and chemical identification, and molecular identification [42]. One of the traditional identification methods, thin-layer chromatography, can distinguish *K. praeruptora*, *Angelica decursiva* (Miquel) Franchet & Savatier, and *S. divaricata* through the number of thin-layer fluorescent spots, color, and Rf value [43]. Fourier Transform Infrared Spectroscopy (FTIR) [44] can not only quickly identify *K. praeruptora* and its counterfeit products but also speculate that there are differences in the composition and content of organic ester compounds, aromatic compounds, and glycoside compounds contained according to the map information [45]. Handheld near-infrared spectroscopy (NIRS) combined with chemical pattern recognition techniques is employed to identify *K. praeruptora* and its counterfeit products based on absorbance in the spectral region of 1405–2442 nm and standard normal variate (SNV) pretreatment [46]. In addition, the utilization of headspace-gas chromatography-ion mobility spectrometry (HS-GC-IMS) in combination with principal component analysis (PCA) [47] and other multivariate analysis methods to examine the volatile components of *K. praeruptora* during different harvest periods is an efficient and quantifiable approach for creating a fingerprint atlas, which aids in the identification of plants [48].

Modern molecular DNA barcoding technology is widely used in the identification of *K. praeruptora* and counterfeit species [49,50]. The use of random amplified polymorphic DNA (RAPD) and inter simple sequence repeat (ISSR) genetic diversity analysis, as well as internally transcribed spacer (ITS) sequencing [8,51,52], allow for the detection of interspecies differences and molecular analysis of botanicals. First, according to the RADP polymorphism of the genomic DNA, at the threshold of 0.66, *A. decursiva* and *K. praeruptora* were divided into two categories, and there were also obvious differences in the amplified bands [53]. Second, the ISSR molecular marker technique can be used to analyze the genetic diversity of *K. praeruptora* from different origins, and it was found that the percentage of polymorphic bands was 95.1% and the genetic similarity coefficient was between 0.201–0.822 [54]. In addition, due to the wide distribution of test sequence information sites and variation sites [55,56,57], the ITS sequence is an ideal DNA barcode to identify *K. praeruptora*, *A. decursiva*, and adulterants [58]. The specific PCR primer pairs QH-CP19s/QH-CP19a and ZHQH-CP3s/ZHQH-CP3a were designed for *K. praeruptora* and A. decursiva, respectively, which was a quick and effective method to distinguish between them [59]. DNA barcoding technology based on ITS4 and ITS5 sequences can successfully identify 13 regional substitutes and 23 adulterants of *K. praeruptora* [8]. Some scholars have also found a specific primer region for identifying camphor in *A. sylvestris*, *K. praeruptora*, and *Anthriscus sylvestris* (L.) Hoffm. through the comparison of rDNA-ITS sequences, and the primer region has produced a 273 bp camphor-specific DNA chain to A. sylvestris. As a result of the RAPD analysis, a Sequence Characteristic Amplified Region (SCAR) marker is developed, and PCR combined with primers can be used to establish a SCAR marker that could simultaneously identify the three plants [54]. The advent of DNA molecular identification technology has led to the discovery of numerous potential DNA barcodes, thereby providing additional feasible identification sites for the species identification of *K. praeruptora*.

With the development of high-throughput techniques, the use of complete chloroplast genomes as one super-barcode should bring better resolution effects than the use of one or even several universal or specific DNA barcodes [60,61]. The complete chloroplast genome of *K. praeruptora* was generated using high-throughput sequencing [62]. Comprehensively comparative and phylogenomic analyses of plastid genomes for seven Peucedanum plants including *K. praeruptora* were performed; the results showed both conservation and diversity and supported the thesis that Peucedanum is not a monophyletic group [14]. These findings can serve as a basis for future studies on the taxonomy and evolution of Peucedanum.

## 6. Phytochemistry

The main chemical components of *K. praeruptora* include coumarin, volatile oil, phenanthrenequinone, organic acids, and sterols [63,64,65]. Coumarin was first isolated from the legume *Dipteryx ordorata* (Aubl.) Forsyth f., which can be found in many plants as secondary metabolites of the roots [66,67], stems [68], leaves [69], flowers, fruits, and seeds. Coumarins were reported to exhibit tremendous biological activities, such as anti-inflammatory, antioxidant, and antidiabetic actions. According to related reports, more than 70 kinds of coumarin compounds have been isolated in the root of *K. praeruptora* [70]. 

Modern pharmacological studies have found that the most important pharmacodynamic components of *K. praeruptora* are divided into simple coumarin, furanocoumarin and pyranocoumarin compounds [71,72,73]; the structural formula of some compounds is displayed in Figure 4. Phenylalamine ammonia lyase (PAL) in plants can convert L-phenylalanine into trans-cinnamic acid, which is activated by cinnamic acid 2-hydroxylase (cinnamic acid under the action of 2-hydroxylase, C2H) and cinnamic acid 4-hydroxylase (cinnamic acid under the action of 4-hydroxylase, C4H). The ortho-hydroxylation and para-hydroxylation reactions occur, which convert it into o-coumaric acid and p-coumaric acid. The p-hydroxylation reaction of trans-cinnamic acid is prior to the ortho-hydroxylation reaction [74]. If there is a lack of C4H in the chloroplast, C2H can convert cinnamic acid into o-coumaric acid [75]. The simple framework of the coumarin biosynthetic pathway is shown in Figure 5 [76,77,78].

### 6.1. Coumarins

According to literature reports, various methods have been employed for the qualitative analysis of chemical components in *K. praeruptora*. These methods include Nuclear Magnetic Resonance spectroscopy (NMR) [79], Mass Spectrometry (MS) [80], and Gas Chromatography-Mass Spectrometry (GC-MS) [81]. Additional techniques such as High-Performance Liquid Chromatography-Ultraviolet Detection (HPLC-UV), Nuclear Magnetic Resonance Spectroscopy (NMRS), and High-Performance Liquid Chromatography-Mass Spectrometry (HPLC-MS) have also been employed [82,83,84] (Table 1). Scholars first discovered two simple coumarin compounds, namely skimmin and scopolin, within *K. praeruptora* in 1989 [79]. To date, a total of eleven simple coumarin compounds have been identified from *K. praeruptora*, as listed in Table 1 (1–11), and some of these coumarins exhibit antitumor properties. For instance, scopolamine activates NF-κB and caspase-3, and resveratrol enhances the activity of Phase II enzymes glutathione S-transferase (GST) and quinone reductase (QR) while also inhibiting the expression of matrix metalloproteinases (MMP-7, -2 and MMP-9), aiming to achieve an antitumor effect [85]. Since the 1990s, researchers have conducted more systematic investigations into the active components of *K. praeruptora* [86,87]. The results indicate that coumarins are the main active constituents of *K. praeruptora*, and, until now, a total of 19 linear furanocoumarins and six angular furanocoumarins have been discovered (Table 1 (12–36)). From the 1970s to 1980s, researchers made the discovery that the primary active component of *K. praeruptora* is praeruptorin A (Pra-A), which has the highest content [88]. They also identified three other saponins derived from *K. praeruptora*: praeruptorin B (Pra-B), praeruptorin C (Pra-C), and praeruptorin D (Pra-D) [89,90]. Until now, more than 15 pyranocoumarin compounds have been separated from *K. praeruptora* (Table 1 (37–52)).

### 6.2. Other Constituents

*K. praeruptora* contains various components, including coumarin compounds, flavonoids, fatty acids, benzoic acids, and steroids. Flavonoid compounds were discovered in *K. praeruptora* for the first time, which were identified as 4H-1-benzopyran-4-one, 5-hydroxy-6-methoxy-2-phenyl-7-O-α-D-glucuronyl methyl ester and 4H-1-benzopyran-4-one, 5-hydroxy-6-methoxy-2-phenyl-7-O-α-D-glucuronyl acid [91]. Additionally, the volatile oil content in *K. praeruptora* is relatively high, which consists of alkanes, esters, ketones, and sesquiterpenes (Table 1 (53–70)). Among them, α-pinene, hinokitol, aromadendrene, terpinolene, α-farnesene, and longifolene account for more than 60% of the relative composition of the volatile oil in *K. praeruptora* [92]. These components, being the main constituents, can also serve as quality control standards for the oil’s analysis and assessment.

**Table 1 molecules-28-08153-t001:** Main chemical compounds isolated from *K. praeruptora*.

Classification	No.	Chemical Components	Formula	References	Test Methods	CAS Identifiers
Simple coumarin	1	Apiosylskimmin	C_20_H_24_O_12_	[26]	UPLC-HR-Orbitrap-MS	103529-94-8
	2	Eleutheroside B1	C_17_H_20_O_10_	[93]	NMR/MS	16845-16-2
	3	Hymexelsin	C_21_H_26_O_13_	[94]	HPLC	117842-09-8
	4	Isofraxidin	C_11_H_10_O_5_	[95]		486-21-5
	5	Isoscopoletin	C_10_H_8_O_4_	[96]	LC-MS	776-86-3
	6	Scopoletin	C_10_H_8_O_4_	[96]	LC-MS	92-61-5
	7	Scopolin	C_16_H_18_O_9_	[97]	UPLC-Q-TOF-HRMS	531-44-2
	8	Skimmin	C_15_H_16_O_8_	[97]	HPLC	93-39-0
	9	Umbelliferone	C_9_H_6_O_3_	[77]	DMR	93-35-6
	10	8-carboxy-7-hydroxy coumarin	C_11_H_8_O_5_	[98]	NMR	5112-55-0
	11	(−)-peucedanol	C_15_H_18_O_5_	[99]	HPLC/PAD	28095-18-3
Furanocoumarin compounds	12	Angelicin	C_11_H_6_O_8_	[80]	HPLC/NMR/MS	523-50-2
	13	Arnocoumarin	C_14_H_10_O_3_			11037-15-3
	14	Apterin	C_20_H_24_O_10_	[100]	LC-MS	53947-89-0
	15	Bergapten	C_12_H_8_O_4_	[101]	HPLC	484-20-8
	16	Imperatorin	C_16_H_14_O_4_	[102]	IMP	482-44-0
	17	Isopimpinellin	C_13_H_10_O_5_	[103]	HPLC	482-27-9
	18	Isorutarin	C_16_H_21_NO_3_	[79]	NMR	53846-51-8
	19	Marmesin	C_14_H_14_O_4_	[104]	NMR/MS	13849-08-6
	20	Marmesinin	C_20_H_24_O_9_	[79]	NMR	27497-13-8
	21	Marmesin-11-O-β-D-glucopyranosyl (1→6)-β-D-glucopyranoside		[104]	NMR/MS	
	22	Nodakenetin	C_14_H_14_O_4_	[89]	NMR/MS	495-32-9
	23	Nodakenetin tiglate	C_19_H_20_O_5_	[105]	HPLC	106974-21-4
	24	Nodakenin	C_26_H_34_O_14_	[106]	UPLC	495-31-8
	25	Oroselol	C_14_H_12_O_4_	[90]	HPLC	1891-25-4
	26	Oxypeucedanin	C_16_H_14_O_5_	[91]	NMR	737-52-0
	27	Oxypeucedanin hydrate	C_16_H_16_O_16_	[91]	NMR	133164-11-1
	28	Peucedanoside A	C_20_H_22_O_10_	[107]	NMR	946122-87-8
	29	Peucedanoside B		[107]	NMR	
	30	Praeroside I		[79]	NMR	
	31	Praeroside VII		[108]	NMR	
	32	Psoralen	C_11_H_6_O_3_			66-97-7
	33	Qianhucoumarin G		[109]	NMR	
	34	Rutarin	C_20_H_24_O_10_	[79]	NMR	20320-81-4
	35	Sphondin	C_12_H_8_O_4_	[110]	HPLC	483-66-9
	36	Xanthotoxin	C_12_H_8_O_4_	[101]	HPLC	298-81-7
Pyranocoumarin compounds	37	Decursinol angelate	C_19_H_20_O_5_	[82]	UHPLC-MS/MS	130848-06-5
	38	Decursitin D	C_19_H_20_O_6_	[111]	HPLC	245446-61-1
	39	d-Laserpitin	C_19_H_20_O_6_	[112]	UPLC-MS/MS	134002-17-8
	40	Longshengensin A	C_21_H_22_O_7_	[113]	NMR/MS	
	41	Marmesin	C_14_H_14_O_4_	[97]	UPLC-Q-TOF-HRMS	13849-08-6
	42	Peucedanocoumarin II	C_21_H_22_O_7_	[114]	NMR	130464-56-1
	43	Praeruptorin A	C_21_H_22_O	[115]	HPLC/LC–MS/MS	73069-25-7
	44	Praeruptorin B	C_24_H_26_O	[73]	HPLC/LC–MS/MS	81740-07-0
	45	Praeruptorin C	C_24_H_28_O_7_	[116]	HPLC/LC–MS/MS	72463-77-5
	46	Praeruptorin D	C_24_H_26_O	[117]	HPLC	
	47	Praeroside II	C_20_H_24_O_10_	[118]	UPLC-HR-Orbitrap-MS	86940-46-7
	48	Pteryxin	C_21_H_22_O_7_	[119]	NMR	13161-75-6
	49	Selinidin	C_19_H_20_O_5_	[74]	UPLC-MS/MS	19427-82-8
	50	Suksdorfin	C_21_H_24_O_7_	[120]	HPLC	53023-17-9
	51	(+)-samidin	C_21_H_22_O_7_	[74]	HPLC	477-33-8
Other constituents	52	Acetylatractylodinol	C_15_H_12_O_3_	[111]	NMR	61582-39-6
	53	Adenosine	C_10_H_13_N_5_O_4_	[121]	NMR	58-61-7
	54	Aromadendrene	C_15_H_24_	[122]	GC/MS	109119-91-7
	55	Baihuaqianhuoside	C_6_H_12_O_6_	[86]	NMR	155969-61-2
	56	Butyric acid	C_4_H_8_O_2_	[93]	NMR	107-92-6
	57	Daucosterol	C_35_H_60_O_6_	[87]	NMR	74-58-8
	58	Galactitol	C_6_H1_4_O_6_	[87]	NMR	608-66-2
	59	Gallic acid	C_7_H_6_O_5_	[86]	NMR	149-91-7
	60	Hinokitol	C_10_H_12_O_2_	[123]	GC/MS	499-44-5
	61	Longifolene	C_15_H_24_	[124]	GC/MS	475-20-7
	62	Mannitol	C_6_H_14_O_6_	[125]	HPLC	69-65-8
	63	Palmitic acid	C_16_H_32_O_2_	[126]	NMR	57-10-3
	64	Tanshinone I	C_18_H_12_O_3_	[93]	NMR/MS	568-73-0
	65	Tanshinone II_A_	C_19_H_18_O_3_	[93]	NMR/MS	568-72-9
	66	Tetracosanoic acid	C_24_H_48_O_2_	[126]	NMR/MS	302912-17-0
	67	Terpinolene	C_10_H_16_	[127]	HS-SPME/GC-MS	586-62-9
	68	Vanillic acid	C_8_H_8_O_4_	[86]	NMR	121-34-6
	69	α-farnesene	C_15_H_24_	[128]	HS-SPME/GC-MS	502-61-4
	70	α-Pinene	C_10_H_16_	[129]	HS-SPME/GC-MS	7785-26-4
	71	β-sitosterol	C_29_H_50_O	[87]	NMR	
	72	2,6-dimethyl quinoline	C_11_H_11_N	[126]	NMR/MS	877-43-0
	73	4H-1-benzopyran-4-one,5-hydroxy-6-methoxy-2-phenyl-7-O-α-D-glucuronyl acid		[91]	Reverse-phase silica gel column	
	74	4H-1-benzopyran-4-one,5-hydroxy-6-methoxy-2-phenyl-7-O-α-D-glucuronyl methyl ester		[91]	Reverse-phase silica gel column	
	75	9,10-dihydrophenanthrinic acid		[130]	NMR	
	76	(−)-sclerodin	C_18_H_16_O_6_	[126]	NMR	104855-18-7

## 7. Pharmacology

*K. praeruptora* represents a gentle Chinese herbal remedy known for its diverse pharmacological effects and demonstrates positive therapeutic efficacy against certain diseases (Table 2). Its main functions include eliminating phlegm and relieving asthma, providing cardiovascular protection, antibacterial and anti-oxidation effects, as well as anti-tumor and anti-diabetes functions (Figure 6).

### 7.1. Eliminate Phlegm and Relieve Asthma

According to the 2020 edition of the Chinese Pharmacopoeia, *K. praeruptora* is recorded as an effective treatment for congested chest and reducing phlegm, dispelling wind, and clearing heat. It is commonly used to alleviate symptoms such as coughing with phlegm-heat, thick yellow phlegm, and wind-heat cough with excessive phlegm. In a designed study, the honey moxibustion treatment involving *K. praeruptora* was administered to mice [130]. The results demonstrate the potent expectorant and antitussive effects of this treatment, which are evaluated based on the release of phenol red. The findings indicate that the combination of *K. praeruptora* with honey moxibustion has the potential to effectively reduce coughing and facilitate expectoration. This is evident through a significant reduction in coughing and improvement in expulsion of respiratory secretions in the treated mice. Moreover, the study reveals that phenol red has a significant impact on coughing in mice. It increased the output of phenol red and inhibited coughing caused by ammonia, while also prolonging the incubation period of coughing. Additionally, phenol red exhibited similar effects on asthma induced by histamine phosphate in guinea pigs. Notably, these effects were found to be stronger than those observed with *K. praeruptora* alone prior to moxibustion with honey. Furthermore, a specific compound called angle-type pyranocoumarin, found in the raceme of Pra-C, was found to inhibit the contraction of rabbit tracheal smooth muscle induced by acetylcholine and potassium chloride (KCl). It particularly demonstrated efficacy in the contraction and relaxation induced by high potassium [158].

### 7.2. Anti-Inflammatory and Analgesic Properties

Previous literature reports suggest that Pra-A inhibits the inflammatory response of macrophages induced by lipopolysaccharide (LPS). It is speculated that this compound exerts anti-inflammatory effects in vitro by inhibiting the activation of the NF-kB signaling pathway [138]. On the other hand, Pra-C has been shown to protect neurons from excitotoxicity by down-regulating glutamate N-methyl-D-aspartate (NMDA) [153]. Additionally, it can inhibit the inflammatory response of RAW264.7 cells stimulated by LPS by blocking the NF-kB and signal transducer and activator of transcription 3 (STAT3) signaling pathways [140]. Therefore, recent studies have suggested that Pra-C may possess anti-inflammatory effects in the central nervous system, particularly on microglia. These findings raise the possibility of utilizing Pra-C for the treatment of chronic pain associated with systemic inflammation.

### 7.3. Cardiovascular Protection

The extract of *K. praeruptora* has been found to improve left ventricular diastolic function, enhance blood supply to the body, and treat heart failure [142]. This improvement is primarily attributed to the important role of two components, Pra-A and Pra-C. Pra-A acts as a calcium blocker and potassium channel opener, offering myocardial protection. On the other hand, Pra-C exhibits various beneficial effects such as anti-myocardial ischemia, anti-heart failure [159], vasodilation, blood pressure reduction [143], inhibition of calcium influx [144], and reduction in myocardial oxygen consumption [145]. Furthermore, studies have shown that the extract derived from *K. praeruptora* can modulate gene expression associated with apoptosis in cardiomyocytes induced by abdominal aortic stenosis. This modulation inhibits myocardial remodeling and offers a biological therapeutic impact on heart failure [141]. Notably, the extract also effectively impedes the expression of c-Jun protein, a crucial nuclear transcription factor involved in the c-Jun N-terminal kinase (JNK) pathway of cell signal transduction, thereby protecting the myocardium. Additionally, research has identified coumarin 7, extracted from the rhizome of *K. praeruptora*, as an inhibitory compound against the proliferation of rat aortic vascular smooth muscle cells. Further investigation and biological testing have revealed that ostrutin7, a coumarin derivative, is the principal anti-proliferation compound that can ameliorate atherosclerosis [144].

Existing studies have demonstrated that Pra-A can effectively decrease mean pulmonary artery pressure, total pulmonary resistance, and pulmonary vascular resistance. Furthermore, it can increase oxygen transport capacity, reduce pulmonary artery vascular resistance, and simultaneously enhance cardiac volume, mixed venous oxygen partial pressure, and oxygen transport capacity [160]. These findings suggest that Pra-A has selectively pulmonary vasodilatory properties, leading to the reduction in pulmonary arterial pressure, pulmonary vascular resistance, and the improvement of cardiac function and tissue oxidation. As a result, it may be utilized as a potential treatment for pulmonary hypertension [141]. In addition, Pra-C has been shown to have blood pressure-lowering effects. Long-term use of Pra-C may reduce blood pressure by increasing the concentration of vascular endothelial nitric oxide (NO). Moreover, Pra-C may also protect against myocardial hypertrophy, ischemia, and an increase in intracellular calcium ions [139]. It can upregulate phospholamban gene expression and Ca^2+^ concentration, which play a crucial role in ensuring normal cardiac function [161]. By inhibiting Ca^2+^ influx and increasing Ca^2+^ uptake in sarcoplasmic reticulum, Pra-C can reduce coronary artery spasm, lower afterload and blood pressure, and protect cardiomyocytes from ischemia-reperfusion injury. Interestingly, the expression of the phospholamban gene in spontaneously hypertensive (SHR) rats was found to be lower than that in normal sprague-dawley (SD) rats, while Pra-C supplementation could up-regulate the expression of the phospholamban gene. This suggest that the low expression of cardiac phospholamban mRNA in SHR may not only be caused by hypertension but also be influenced by mechanisms unrelated to blood pressure [133,162]. Overall, Pra-C has the potential to regulate calcium levels and lower blood pressure and may serve as a promising therapeutic agent for hypertension and related cardiovascular diseases. 

### 7.4. Antineoplastic Activity

Pra-A and Pra-B exhibit anti-proliferation and cytotoxic effects against human gastric cancer cells [147]. Pra-A has demonstrated the ability to enhance the efficacy of doxorubicin (DOX) in treating gastric cancer cells, leading to a more pronounced reduction in cell growth when used in combination with DOX compared to DOX treatment alone. Notably, Pra-B shows promise as an anti-metastasis agent for renal cell carcinoma [149]. It achieves this by inhibiting the epidermal growth factor receptor extracellular-regulated kinase/mitogen-activated protein kinase (EGFR-MEK-ERK) signaling pathway, which consequently down-regulates cathepsin C (CTSC) and cathepsin V (CTSV). As a result, the migration and invasion abilities of human renal cancer cells are significantly decreased.

### 7.5. Other Pharmacological Effects

According to a previous study, the volatile oil samples of *K. praeruptora* extracted with 95% ethanol and petroleum ether at 60 °C to 90 °C demonstrated positive effects on Escherichia coli, Staphylococcus aureus, Bacillus typhi, and Shigella flexneri, exhibiting certain antibacterial or bactericidal properties [163]. Additionally, it was reported that Pra-A and Pra-C can significantly up-regulate the expression of multidrug resistance-associated protein 2 through the combined androgen receptor-mediated pathway, providing novel therapeutic approaches for the related cancers [152]. Furthermore, the chemical components of *K. praeruptora* have been identified using an HEK293 α1A AR cell membrane chromatography (HEK293 α1A/CMC) column and UPLC-ESI-MS/MS coupling system, revealing that Pra-A, Pra-B, and Pra-C exhibit α1 adrenergic receptor activity [155].

Furthermore, to combat the coronavirus disease 2019 (COVID-19) pandemic and showcase the benefits of traditional Chinese medicine in pre-treatment, Jilin Provincial Hospital of Traditional Chinese Medicine has taken proactive measures. The hospital has leveraged traditional Chinese medicine and prepared over 50,000 decoctions, mainly composed of traditional Chinese medicines such as *Lonicera japonic*a Thunb, *K. praeruptora*, *Glycyrrhiza uralensis* Fisch, *Stemona japonica* (BI.) Miq, *Perilla frutescens* (L.) Britt., *Ephedra sinica* Stapf, and raw gypsum. It has the effects of treating congested chest and strengthening the exterior, clearing away heat and detoxification, clearing the lung and relieving the throat, and is beneficial to many people in the city [164]. The key groups in each isolation point will be intervened with traditional Chinese medicine to help them improve their immunity and scientifically assist in the prevention and control of the epidemic. Modern pharmacological studies have found that the most important pharmacodynamic components of *K. praeruptora* are dihydropyranoid coumarin compounds, including Pra-A, Pra-B, and Pra-C [9,91], especially Pra-A and Pra-B, which were included in the quality evaluation index of the 2020 edition of Chinese Pharmacopoeia.

## 8. Safety and Toxicity

General pharmacological studies revealed that oral administration of Pra-A and Pra-B did not induce behavioral effects or acute toxicity at a dose of 1 g/kg [150]. Furthermore, cytotoxicity tests on RAW264.7 cells and primary mouse peritoneal macrophages showed that Pra-A did not affect cell viability in the dose range of 1 to 100 mg/mL. Similarly, Pra-B did not exhibit cytotoxic effects in primary mouse peritoneal macrophages at doses ranging from 1 to 60 mg/mL, indicating a favorable safety profile in vitro. Nevertheless, further evaluation of the in vivo safety profile is warranted [165]. Acute toxicity tests in laboratory animals demonstrated a wide margin of safety with no observed adverse effects at recommended therapeutic doses. However, research showed that when administered to mice separately via oral gavage and intraperitoneal injection, methanol extract of *K. praeruptora* exhibited lower toxicity when administered orally, whereas intraperitoneal injection resulted in higher toxicity. This finding should raise caution in clinical drug administration [166]. In the investigation of the extracellular anti-tumor activity of extracts from *K. praeruptora*, higher concentrations of the extract treatment resulted in a decreased proliferation rate of human hepatocellular carcinomas (HepG2 cells). However, it is important to note that the cytotoxicity of the chemotherapeutic drug doxorubicin against HepG2 cells was significantly greater than that of *K. praeruptora* extracts [167].

## 9. Conclusions and Perspectives

*K. praeruptora*, a renowned traditional Chinese herbal medicine with diverse chemical components and extensive pharmacological effects, exhibits efficacy in treating conditions like wind-heat headache, phlegm-heat cough, and asthma. Its cardiovascular benefits, cancer prevention properties, and antimicrobial activities contribute to its prominence in ethnopharmacology. Despite its growing cultivation in China, addressing knowledge gaps is crucial.

First, cultivation systems need improvement, necessitating a deeper understanding of the bolting gene and mechanism. Current research lacks systematicity, hindering practical advancements. Second, comprehensive research into *K. praeruptora*’s chemical components, particularly volatile oils, phenanthrenequinone, and sterols, is essential. Utilizing advanced analytical techniques can enhance understanding. Third, physiological and molecular studies should focus on stress responses and the relationship between microorganisms and secondary metabolites. Whole-genome sequencing is imperative to address research gaps.

Regarding cardiovascular effects, research is extensive, but broader physiological impacts remain underexplored. Leveraging biotechnology for active constituent extraction holds promise. Plant-derived microRNAs also play a vital role in human health, influencing growth, development, and gut microbiota [168,169,170,171]. Acknowledging this in the context of *K. praeruptora* enhances its development of drug mechanisms. Lastly, while *K. praeruptora* shows potential for disease prevention and treatment, further clinical trials are essential. Overcoming obstacles such as traditional usage, complex composition, and cultural barriers will promote its integration into modern healthcare practices. Future research should focus on bolting characteristics, chemical composition, and bioinformatics to unravel its mechanisms and expand medical applications, solidifying its economic and medical significance.

## Figures and Tables

**Figure 1 molecules-28-08153-f001:**
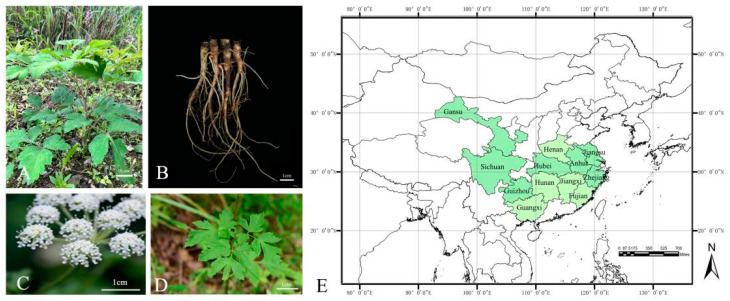
The overall appearance (**A**), roots (**B**), flowers (**C**), blades (**D**) view, and distribution (**E**) of *K. praeruptora*. (The green shading in E represents the distribution of *K. praeruptora*).

**Figure 2 molecules-28-08153-f002:**
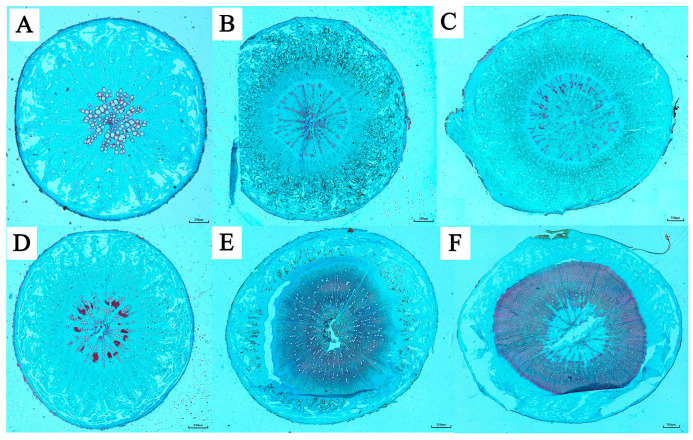
Anatomical differences between (**A**–**C**) unbolting and (**D**–**F**) bolting plants of *K. praeruptora* at different growth stages [23]. The unbolting roots’ parenchyma cells, initially undifferentiated, appear large and loose (**A**). The pericycle differentiates into conduits, woody tissue, and wood rays (**B**). Woody tissue parenchyma cells are mildly fibrose, and secretion ducts increase (**C**). Bolting plants show earlier pericycle differentiation and woody parenchyma fibrosis (**D**). Pericycle shrinks, cracks increase, secretion ducts decrease, and fibrosis intensifies, leading to pith formation in later stages (**E**,**F**).

**Figure 3 molecules-28-08153-f003:**
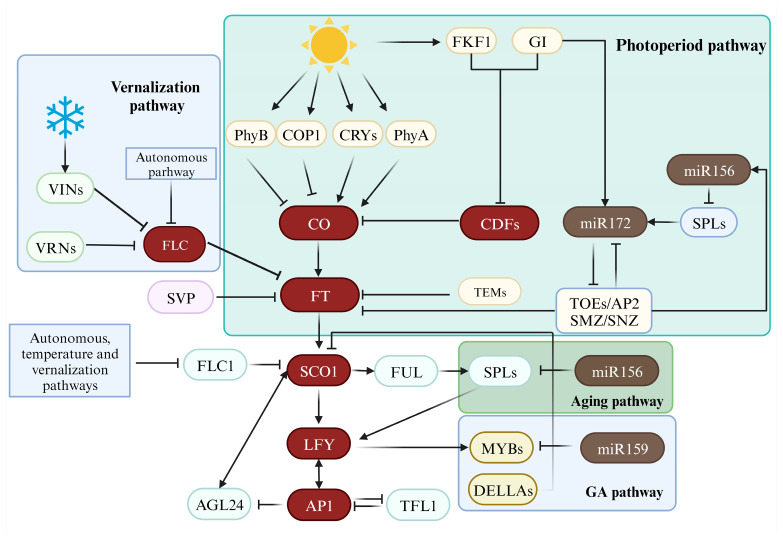
Regulatory pathways of plant bolting and flowering.

**Figure 4 molecules-28-08153-f004:**
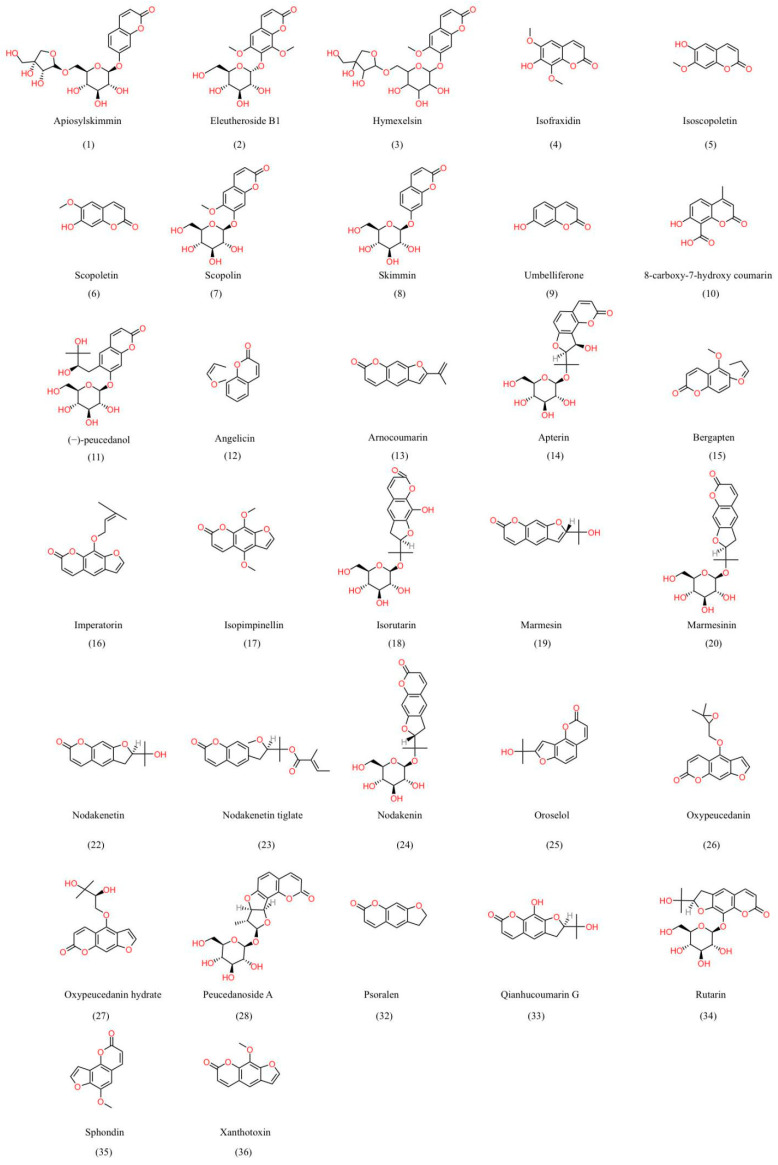
The structures of chemical constituents isolated from *K. praeruptora*.

**Figure 5 molecules-28-08153-f005:**
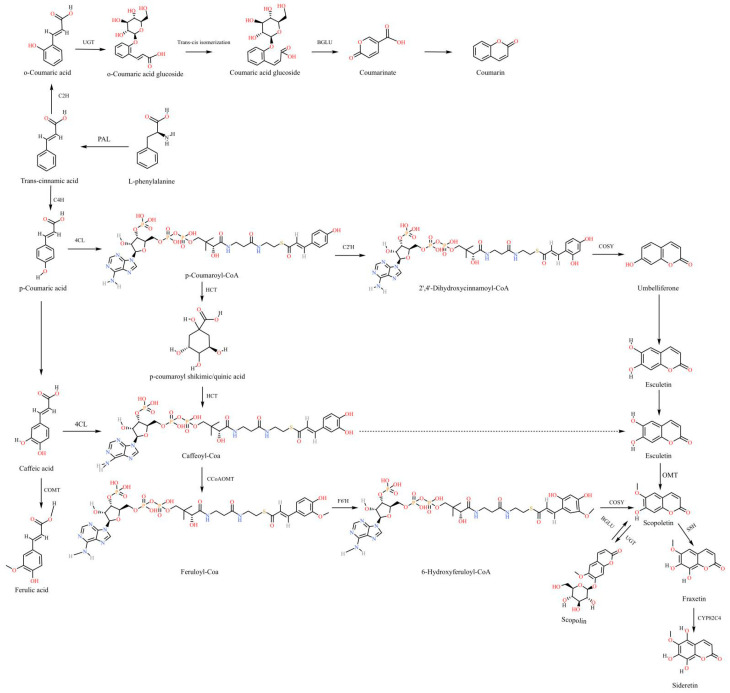
A simple route to the synthesis of coumarin.

**Figure 6 molecules-28-08153-f006:**
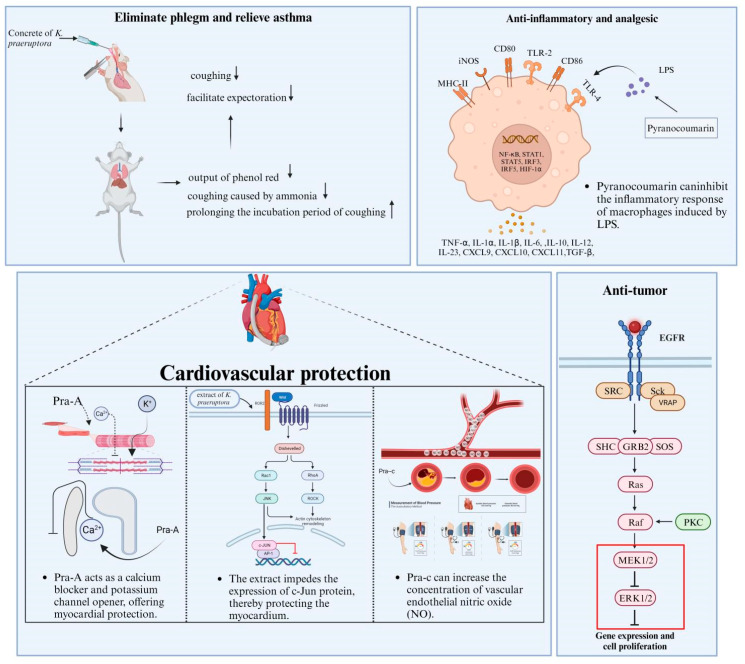
The pharmacological properties of *K. praeruptora*.

**Table 2 molecules-28-08153-t002:** Pharmacological activities of *K. praeruptora* (“↑“ Upward arrow indicates an increase in expression levels. “↓” Downward arrow indicates a decrease in expression levels.).

Pharmacological Activity	Compounds/Extracts	Material/Model	Treatment	Results	References
Eliminate phlegm and relieve asthma	Praeruptorin A	Rabbits/trachea extricated and immersed in a nutritive solution containing a mixed gas	In vivo 30 µmol/L 0.5 h	↓ Ca^+^ and PDC	[131]
		Mice/chronic asthma model	In vivo 30, 60, 120 mg/kg 5 d	↓ (IL)-4, IL-5, IL-13, LTC4 and (Ig) E, eotaxin protein and mRNA expression	[132]
		Mice/chronic asthma model	In vivo 30, 60, 120 mg/kg 56 d	↓ IL-4 and IL-13 in BALF, IgE, TGF-β1 and pSmad2/3 expression	[133]
	Honey-roasted *K. praeruptora* pieces	Mice/histamine phosphate asthma model	In vivo 2, 4, 8 g/kg 3 d	↑ Secretion of phenol red	[130]
	Coumarin constituent	Rabbits	In vivo 2, 5, 10 mg/mL	↓ KCl and acetylcholine level	[134]
Anti-inflammatory and analgesic	TCP	Mice	In vivo 31.3, 62.5, 125, 250, 500 µg/mL	↓ MDA	[135]
	Praeruptorin A, B, and E	Rat hepatocytes	In vivo 1 nmol/L	↓ NO synthase expression ↓ mRNAs encoding	[136]
	Praeruptorin A	Croton oil-induced ear dermatitis in mice	In vivo 0.3 μmol/cm^2^	↓ 22–43% oedema reduction	[137]
		Macrophage cells	In vitro 6.25, 12.5, 25 mg/mL 24 h	↓ NO, IL-1β and TNF-α, mRNA expressions, iNOSand TNF-α	[138]
		Mice/chronic asthma model	In vivo 30, 60, 120 mg/kg 5 d	↓ (IL)-4, IL-5, IL-13, LTC4 and (Ig) E, eotaxin protein and mRNA expression	[132]
		Mice/chronic asthma model	In vivo 30, 60, 120 mg/kg 56 d	↓ IL-4 and IL-13 in BALF, IgE, TGF-β1 and pSmad2/3 expression	[133]
	Praeruptorin C	Cortical neuron cells	In vitro 0, 0.1, 1, 10 µmol/L	↓ NMDA and Bcl-2	[139]
	Praeruptorin C, D, and E	Macrophage cells	In vitro 4, 8, 16 μg/mL 18 h	↓ NO, IL-6 and TNF-α, mRNA and r κB-α protein expression	[140]
Cardiovascular protection	Praeruptorin A	Open-chest anesthetized rats	In vivo 2 mg/mL 18 h	↓ IL-6, Fas, bax and bcl-2 level	[141]
	Praeruptorin C	Spontaneously hypertensive rats	In vivo 20 mg/kg 48 d	↓ PLB mRNA expression ↑ HMI and LVMI	[142]
		Renovascular hypertensive rats/2K1C-RHR	In vivo 20 mg/kg 63 d	↑CF/HWW, CO/HWW, LVSP, -dp/d t_max_ value ↓ LVEDP and T value	[143]
		Renovascular and spontaneously hypertensive rats	In vivo 20 mg/kg 63 d	↑ Ca^2+^ and NO level	[144]
		Cortical neuron cells	In vitro 0, 0.1, 1, 10 µmol/L	↓ NMDA and Bcl-2	[140]
	Praeruptorin A, B, C, and D	Heart hypertrophy rats/2K1C-RHR	In vivo 30 mg/kg 63 d	↑-dp/d t_max_ value ↓ LVEDP and HYP level	[145]
	Ostruthin	Vascular smooth muscle cells	In vitro 1, 3, 10, 30 µmol/L 20 h	↓ VSMC proliferation concentration	[146]
Anti-tumor effect	Methanolic extract	SGC7901 gastric cancer cells	In vitro 50, 100, 200, 300 µg/mL 24 h	↑ LDH ↓ Cell proliferation	[147]
	Praeruptorin A	HeLa and SiHa cells	In vitro 0, 10, 20, 30, 40, 50 µmol/L 24 h	↓ MMP-2, cyclin D1, Skp2, ERK1/2 ↑ TIMP-2, Rb, p16, p21, p27,	[148]
	Praeruptorin B	Renal cell carcinoma	In vitro 0, 10, 20, 30, 40, 50 µmol/L 24 h	↓ EGFR, MEK, ERK, CTSC, CTSV, mRNA and protein expression	[149]
		HeLa cells and C3H10T1/2 cells	In vitro 0, 10, 20, 30, 40, 50 µmol/L 140 d	↓ Tumor promotion	[150]
	Praeruptorin C	Human lung cancer cell lines	In vitro 0, 10, 20,30 µmol/L 24 h	↓ cyclin D, cathepsin D, ERK1/2 ↑ p21 protein,	[151]
	Praeruptorin A and C	HepG2 Cells	In vitro 10, 25, 50, 100, 200 µmol/L 18 and 24 h	↓ mRNA and protein expression	[152]
	Praeruptorin D and E	Mice/acute lung injury	In vivo 0, 20, 40, 80 mg/kg 1 h	↓ MPO, IκB-α and p65 protein translocation	[153]
Other pharmacological effects	Praeruptorin C	Bone marrow Monocyte/macrophage (BMM) cells	In vitro 0, 5, 10, 20 µmol/L 48 h	↓ kappa B and c-Jun N-terminal kinase/mitogen-activated protein kinase	[154]
	Praeruptorin A, B, and C	HEK293 cell line	In vitro 1 mg/mL 1 h	Identified with α1A adrenergic receptor activity	[155]
	Terpinolene	Harmful algal blooms (HABs)	In vivo 0.551, 0.881, 1.079, 1.233, 1.470 mmol/L 4 d	↓ NR and GS activities ↑ COX II, ABC transporter, CaBPs expression	[156]
	Marmesinin	Hloroquine-sensitive strain of *P. falciparum* (D10)	In vitro 0.1, 1, 5, 25, and 100 µmol/L 24 h	↓ Antiplasmodial D10 ↑ Cytotoxicity SK-OV-3	[157]

## Data Availability

Not applicable.

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
