# Peer review of "A Review on the Morphology, Cultivation, Identification, Phytochemistry, and Pharmacology of Kitagawia praeruptora (Dunn) Pimenov"

_molecules, 2023, doi:10.3390/molecules28248153_

Round 1

Reviewer 1 Report

Comments and Suggestions for Authors

This review is very interesting as it collects the information about this species of the Apiaceae family that is endemic from China and not widely known. Therefore, the spreading of data about this plant can be useful. However, I have some suggestions for the authors before the paper can be published.

1. In the title, the term "botany-2 should be replaced with "morphology" or "anatomy" or similar..

2. The sentence at line 14-15 should contain the indication that it is referred to the investigated plant species

3. Figure 3, the authors should describe better in the caption what there is in the panels.. sections of what? which part of the plant? what are the tissues represented? Specify for each panel. Moreover the blue color of the sections is not normal.. why this?

4. The chapter 3 should be splitted in two: "Botany" and "Ethnobotany" (not ethnopharmacology as the first includes also the second concept)

5. Figure 6: what does it mean "a simple route"?? Delete it or modify it showing all the biochemical steps.

6. In the conclusions and among the future perspectives the authors should mention the possibility that this plant species contains plant miRNAs with potential cross-kingdom bioactivity on human cells, tissues and organisms. Indeed, it would be interesting to deepen this topic in the future, as recently it has been proposed that not only plant secondary metabolites but also microRNAs from medicinal plants and plant derived foods can exert potent biological function on human health. I would suggest to the authors to see and mention the following works in their paper to support this: BMC Plant Biology. 2023 Sep 19;23(1):439; Journal of agricultural and food chemistry. 2021 Nov 23;69(48):14372-86; PloS one. 2017 Feb 27;12(2):e0172981; Advances in Nutrition. 2021 Jan;12(1):197-211.

7. In figure 1, a dimension bar on the images of plants should be reported.

Author Response

Reviewer 1:

  1. In the title, the term "botany" should be replaced with "morphology" or "anatomy" or similar..

Response: Thank you for your suggestion and reminding. We have replaced the term "botany" with "morphology" in the title and keywords.

  1. The sentence at line 14-15 should contain the indication that it is referred to the investigated plant species.

Response: Thank you for your suggestions. We have modified "this herb" into "K. praeruptora" and highlighted this part in the manuscript.

  1. Figure 3, the authors should describe better in the caption what there is in the panels.sections of what? which part of the plant? what are the tissues represented? Specify for each panel. Moreover the blue color of the sections is not normal. why this?

Response: Thank you for your valuable suggestions. We have incorporated specific descriptions for Figure 3, explaining each part of the image and the reasons for variations in color depth.

  1. The chapter 3 should be splitted in two: "Botany" and "Ethnobotany" (not ethnopharmacology as the first includes also the second concept).

Response: Thank you for the suggestions and explanations you provided. We have splitted the chapter 3 into two: "Botany" and "Ethnobotany".

  1. Figure 6: what does it mean "a simple route"?? Delete it or modify it showing all the biochemical steps.

Response: We have modified figure 6 showing all the biochemical steps.

  1. In the conclusions and among the future perspectives the authors should mention the possibility that this plant species contains plant miRNAs with potential cross-kingdom bioactivity on human cells, tissues and organisms. Indeed, it would be interesting to deepen this topic in the future, as recently it has been proposed that not only plant secondary metabolites but also microRNAs from medicinal plants and plant derived foods can exert potent biological function on human health. I would suggest to the authors to see and mention the following works in their paper to support this: BMC Plant Biology. 2023 Sep 19;23(1):439; Journal of agricultural and food chemistry. 2021 Nov 23;69(48):14372-86; PloS one. 2017 Feb 27;12(2):e0172981; Advances in Nutrition. 2021 Jan;12(1):197-211.

Response: Thank you for your suggestions. We have added a discussion on the potential microRNAs of this plant in the concluding section and cited these four references.

  1. In figure 1, a dimension bar on the images of plants should be reported.

Response: We have added the dimension bars for figure 1.

Reviewer 2 Report

Comments and Suggestions for Authors

Dear Authors,

Reviewer comments molecules-2768647

The manuscript entitled „A review of botany, cultivation, identification, phytochemistry and pharmacology of Kitagawia praeruptora (Dunn) Pimenov“ represents a review on Chinese traditional medicinal plant Kitagawia praeruptora from Apiaceae family containing all necessary information covering its botanical traits, cultivation techniques, phytochemical compositions, ethnopharmacologyand pharmacopeia. An overview of over 70 phytochemicals including coumarins, furanocoumarins, flavonoids extracted from K. praeruptora is provided and their potential medicinal effects are discussed.

I have no major comments on the presented manuscript. I can recommend the manuscript for publication in Molecules.

I have only a few minor comments on the present manuscript which are provided below:

1/ In Figure 1 providing representative photos of K praeruptura plant parts including roots, inflorescence, leaves, etc., and the map of China with indicated sites of K. praeruptora presence, appropriate scale bars have to be added to all photos and the map.

2/ Formal comments on the text related to English language and style:

Title: In the manuscript title, the word „of“ has to be modified to „on“ in the manuscript title: „A review on the botany, cultivation, identification, phytochemistry and pharmacology of Kitagawia praeruptora (Dunn) Pimenov“.

Line 69: Modify the word form „have“ to „has“ in the statement: „K. praeruptora is a perennial herb that grows up to 1 meter in height and has a solitary stem ….“

Line 99: Modify the word form „firstly“ to „first“ in the statement: „It was first recorded in publication by Ming Yie Bie Lu and Ben cao Jing Ji Zhu…“

Line 162: The term „yield“ is used with respect to K. praeruptora usage for medicinal purpose. It has to be specified which part of the plant is harvested for medicinal purposes, i.e., forms the „yield.“

Line 218: Modify the word form „Secondly“ to „Second“ in the statement: „Second, the ISSR molecular marker technique…“

Line 225: Add a comma between the words „A. decursiva“ and „respectively.“

Line 243: Add the words „the thesis that“ in the statement: „…the results showed both conservation and diversity and supported the thesis that Peucedanum is not a monophyletic group…“

Line 267, Table 1 legend: Add the verb „isolated“ in the Table 1 legend, i.e., „Main chemical compounds isolated from K. praeruptora“.

Line 285: Add a comma (twice) in the statment ů…and, till now, a total of 19 linear furanocoumarins…“

Line 340: Add the word „propertiesů in the heading „7.2 Anti-inflammatory and analgetic properties“.

Line 381: Remove extra word „also“ in the statement „Moreover, Pra-C may also protect against myocardial hypertrophy,…“

Line 434, Figure 7 legend: Modify the word „activities“ to „properties“ in the statement „The pharmacological properties of K. praeruptora“.

Line 451. Use the full verb form instead of a contracted form in the statement: „However, i tis important to note that…“

Line 481: Modify the word form „Thirdly“ to „Third“ in the statement „Third, there are several areas…“

Line 491: Modify the word form „Fourthlyů to „Fourth“ in the statement „Fourth, i tis worth noting that…“

Line 500: Remove the word „as“ in the statement „…and safety of a medicinal agent remain unclear.“

Final recommendation: Accept after a minor revision.

Comments on the Quality of English Language

Dear Authors,

Reviewer comments molecules-2768647

The manuscript entitled „A review of botany, cultivation, identification, phytochemistry and pharmacology of Kitagawia praeruptora (Dunn) Pimenov“ represents a review on Chinese traditional medicinal plant Kitagawia praeruptora from Apiaceae family containing all necessary information covering its botanical traits, cultivation techniques, phytochemical compositions, ethnopharmacologyand pharmacopeia. An overview of over 70 phytochemicals including coumarins, furanocoumarins, flavonoids extracted from K. praeruptora is provided and their potential medicinal effects are discussed.

I have no major comments on the presented manuscript. I can recommend the manuscript for publication in Molecules.

I have only a few minor comments on the present manuscript which are provided below:

1/ In Figure 1 providing representative photos of K praeruptura plant parts including roots, inflorescence, leaves, etc., and the map of China with indicated sites of K. praeruptora presence, appropriate scale bars have to be added to all photos and the map.

2/ Formal comments on the text related to English language and style:

Title: In the manuscript title, the word „of“ has to be modified to „on“ in the manuscript title: „A review on the botany, cultivation, identification, phytochemistry and pharmacology of Kitagawia praeruptora (Dunn) Pimenov“.

Line 69: Modify the word form „have“ to „has“ in the statement: „K. praeruptora is a perennial herb that grows up to 1 meter in height and has a solitary stem ….“

Line 99: Modify the word form „firstly“ to „first“ in the statement: „It was first recorded in publication by Ming Yie Bie Lu and Ben cao Jing Ji Zhu…“

Line 162: The term „yield“ is used with respect to K. praeruptora usage for medicinal purpose. It has to be specified which part of the plant is harvested for medicinal purposes, i.e., forms the „yield.“

Line 218: Modify the word form „Secondly“ to „Second“ in the statement: „Second, the ISSR molecular marker technique…“

Line 225: Add a comma between the words „A. decursiva“ and „respectively.“

Line 243: Add the words „the thesis that“ in the statement: „…the results showed both conservation and diversity and supported the thesis that Peucedanum is not a monophyletic group…“

Line 267, Table 1 legend: Add the verb „isolated“ in the Table 1 legend, i.e., „Main chemical compounds isolated from K. praeruptora“.

Line 285: Add a comma (twice) in the statment ů…and, till now, a total of 19 linear furanocoumarins…“

Line 340: Add the word „propertiesů in the heading „7.2 Anti-inflammatory and analgetic properties“.

Line 381: Remove extra word „also“ in the statement „Moreover, Pra-C may also protect against myocardial hypertrophy,…“

Line 434, Figure 7 legend: Modify the word „activities“ to „properties“ in the statement „The pharmacological properties of K. praeruptora“.

Line 451. Use the full verb form instead of a contracted form in the statement: „However, i tis important to note that…“

Line 481: Modify the word form „Thirdly“ to „Third“ in the statement „Third, there are several areas…“

Line 491: Modify the word form „Fourthlyů to „Fourth“ in the statement „Fourth, i tis worth noting that…“

Line 500: Remove the word „as“ in the statement „…and safety of a medicinal agent remain unclear.“

Final recommendation: Accept after a minor revision.

Author Response

Reviewer 2:

  1. In Figure 1 providing representative photos of K praeruptoraplant parts including roots, inflorescence, leaves, etc., and the map of China with indicated sites of praeruptora presence, appropriate scale bars have to be added to all photos and the map.

Response: Thank you for your suggestion. We have added the dimension bars for figure 1.

  1. Formal comments on the text related to English language and style:

Title: In the manuscript title, the word "of" has to be modified to "on" in the manuscript title: "A review on the botany, cultivation, identification, phytochemistry and pharmacology of Kitagawia praeruptora (Dunn) Pimenov".

Line 69: Modify the word form " have" to " has" in the statement: "K. praeruptora is a perennial herb that grows up to 1 meter in height and has a solitary stem …."

Line 99: Modify the word form " firstly" to " first" in the statement: " It was first recorded in publication by Ming Yie Bie Lu and Ben cao Jing Ji Zhu…"

Line 162: The term "yield" is used with respect to K. praeruptora usage for medicinal purpose. It has to be specified which part of the plant is harvested for medicinal purposes, i.e., forms the "yield."

Line 218: Modify the word form "Secondly" to "Second" in the statement: " Second, the ISSR molecular marker technique…"

Line 225: Add a comma between the words "A. decursiva" and " respectively."

Line 243: Add the words "the thesis that" in the statement:"…the results showed both conservation and diversity and supported the thesis that Peucedanum is not a monophyletic group…

Line 267, Table 1 legend: Add the verb "isolated" in the Table 1 legend, i.e., "Main chemical compounds isolated from K. praeruptora".

Line 285: Add a comma (twice) in the statment …and, till now, a total of 19 linear furanocoumarins…"

Line 340: Add the word "properties in the heading "7.2 Anti-inflammatory and analgetic properties".

Line 381: Remove extra word " also" in the statement "Moreover, Pra-C may also protect against myocardial hypertrophy…"

Line 434, Figure 7 legend: Modify the word "activities" to "properties" in the statement "The pharmacological properties of K. praeruptora".

Line 451. Use the full verb form instead of a contracted form in the statement: " However, i tis important to note that…"

Line 481: Modify the word form " Thirdly" to " Third" in the statement " Third, there are several areas…"

Line 491: Modify the word form " Fourthly to " Fourth" in the statement " Fourth, it is worth noting that…"

Line 500: Remove the word " as" in the statement " …and safety of a medicinal agent remain unclear."

Response: Thank you for your reminding. We have made modifications to these English language conventions and styles according to your suggestions, and we have highlighted these changes.

Reviewer 3 Report

Comments and Suggestions for Authors

The review article is excellently written, captivating, and informative. It extensively covers a broad range of research on the respective plant. However, there are a few minor corrections that need to be addressed. Firstly, I recommend the authors to remove point 2, "Methods", and specify in the text which tests are conducted in vitro and in vivo.

Furthermore, there are some technical errors within the text that should be corrected. Additionally, it would be beneficial to review the cited sources to ensure that there are no contradictions.

 - Search strategy: While the authors mention searching various scientific databases, they should provide more information on the specific keywords and search terms used. Including specific search strings and criteria can help ensure that all relevant research is captured. The authors should clearly define the criteria used to include or exclude studies. This can help ensure the selection of high-quality and relevant literature. 

- Reproducibility: The authors should provide transparent and reproducible methods to enable other researchers to replicate their search strategy and replicate the review process. This could include specifying the exact date of the search, the search strings used, and the inclusion and exclusion criteria applied. 

- Further controls: In addition to the wide keywords used for the search, the authors could further improve their methodology by considering additional controls. These may include specifying filters based on the publication date, study design, or the inclusion of only peer-reviewed articles. 

The tables in the article are thoughtfully organized and properly cited. It is important to ensure that the sources of figures are appropriately acknowledged, either through citation of the literary source or by claiming copyright, especially in cases where the article containing the figure is not openly accessible. 

Considering these improvements and further controls can strengthen the methodology of the review and enhance the reliability and validity of the findings. 

  In conclusion, the findings of this review article effectively address the main question posed. However, the length of the conclusion should be reduced for better readability and clarity. By rewriting the conclusion and making it more concise, it can enhance its impact and better communicate the key takeaways from the review article. 

Lastly, the conclusion appears to be overly lengthy and could benefit from a more concise summary.

Author Response

Reviewer 3:

The review article is excellently written, captivating, and informative. It extensively covers a broad range of research on the respective plant. However, there are a few minor corrections that need to be addressed.

Firstly, I recommend the authors to remove point 2, "Methods", and specify in the text which tests are conducted in vitro and in vivo.

Furthermore, there are some technical errors within the text that should be corrected. Additionally, it would be beneficial to review the cited sources to ensure that there are no contradictions.

Search strategy: While the authors mention searching various scientific databases, they should provide more information on the specific keywords and search terms used. Including specific search strings and criteria can help ensure that all relevant research is captured. The authors should clearly define the criteria used to include or exclude studies. This can help ensure the selection of high-quality and relevant literature.

Reproducibility: The authors should provide transparent and reproducible methods to enable other researchers to replicate their search strategy and replicate the review process. This could include specifying the exact date of the search, the search strings used, and the inclusion and exclusion criteria applied.

Further controls: In addition to the wide keywords used for the search, the authors could further improve their methodology by considering additional controls. These may include specifying filters based on the publication date, study design, or the inclusion of only peer-reviewed articles.

The tables in the article are thoughtfully organized and properly cited. It is important to ensure that the sources of figures are appropriately acknowledged, either through citation of the literary source or by claiming copyright, especially in cases where the article containing the figure is not openly accessible.

Considering these improvements and further controls can strengthen the methodology of the review and enhance the reliability and validity of the findings.

In conclusion, the findings of this review article effectively address the main question posed. However, the length of the conclusion should be reduced for better readability and clarity. By rewriting the conclusion and making it more concise, it can enhance its impact and better communicate the key takeaways from the review article.

Lastly, the conclusion appears to be overly lengthy and could benefit from a more concise summary.

Response: Thank you for your suggestions to strengthen the methodology of the review and enhance the reliability and validity of the findings. We have removed point 2, 'Methods' and added the relevant methodology regarding the review strategy. In addition, we have rewritten the conclusion to make it more concise.